# Evaluation of the Effectiveness of Derived Features of AlphaFold2 on Single-Sequence Protein Binding Site Prediction

**DOI:** 10.3390/biology11101454

**Published:** 2022-10-03

**Authors:** Zhe Liu, Weihao Pan, Weihao Li, Xuyang Zhen, Jisheng Liang, Wenxiang Cai, Fei Xu, Kai Yuan, Guan Ning Lin

**Affiliations:** 1School of Biomedical Engineering, Shanghai Jiao Tong University, Shanghai 200030, China; 2State Key Laboratory of Functional Materials for Informatics, Shanghai Institute of Microsystem and Information Technology (SIMIT), Chinese Academy of Sciences, Shanghai 200050, China; 3Shanghai Key Laboratory of Psychotic Disorders, Shanghai 200030, China

**Keywords:** AlphaFold2, deep learning, single-sequence protein binding site prediction, feature engineering, spatial filtering

## Abstract

**Simple Summary:**

With the development of artificial intelligence, researchers can roughly predict the crystal structure of a protein by computer without the need for biological experiments, which provides new ideas and solutions to problems, such as protein-protein interaction and drug-target predictions. In this study, we proposed strategies to combine predicted protein structures with deep learning networks and evaluated them on different protein binding site prediction tasks. Our computational experiment results showed that all proposed strategies could effectively encode structural information for deep learning models.

**Abstract:**

Though AlphaFold2 has attained considerably high precision on protein structure prediction, it is reported that directly inputting coordinates into deep learning networks cannot achieve desirable results on downstream tasks. Thus, how to process and encode the predicted results into effective forms that deep learning models can understand to improve the performance of downstream tasks is worth exploring. In this study, we tested the effects of five processing strategies of coordinates on two single-sequence protein binding site prediction tasks. These five strategies are spatial filtering, the singular value decomposition of a distance map, calculating the secondary structure feature, and the relative accessible surface area feature of proteins. The computational experiment results showed that all strategies were suitable and effective methods to encode structural information for deep learning models. In addition, by performing a case study of a mutated protein, we showed that the spatial filtering strategy could introduce structural changes into HHblits profiles and deep learning networks when protein mutation happens. In sum, this work provides new insight into the downstream tasks of protein-molecule interaction prediction, such as predicting the binding residues of proteins and estimating the effects of mutations.

## 1. Introduction

Despite recent progress in deep learning-based protein structure prediction, there remains a gap between structure and function [1]. The interactions between proteins and their interacting partners, including proteins, nucleic acids, and small molecules, mediate most biological processes [2]. Predicting the binding residues of protein-protein interactions will contribute to designing new antibacterial drugs [3], and RNA-protein binding site prediction can give new insights into the mechanisms behind diseases [4]. Conventional biological experimental methods, such as affinity systems, two-hybrid assay, and affinity purification coupled to mass spectrometry, are labor-intensive and time-consuming [5,6,7,8]. Consequently, many artificial intelligence (AI)-based studies have emerged predicting the binding residues of proteins from sequences with the help of machine learning [3,4,9,10,11]. However, due to the limited number of accurately measured protein crystal structures [12,13], most sequence-based predictors can only be trained using evolutionary conservatism profiles and predicted secondary structure.

AlphaFold2 (AF2) [14] is a DeepMind-released deep learning-based framework that can predict the 3D structure of some proteins with remarkable accuracy from primary sequences. DeepMind also released another deep learning framework, named AlphaFold-Multimer [15], to predict the structure of the complex of proteins. However, when facing single chains, the model does not perform well [15]. Therefore, when dealing with a single-sequence prediction task, a direct consideration is predicting the binding site based on the structure generated by AlphaFold2 (Figure 1A,B).

Some studies have indicated that directly feeding the coordinates (the position of residues in space) of the proteins into the model is not an ideal choice [16,17,18]. One possible reason is that deep learning networks may not be able to directly match the coordinates of atoms to the fold pattern of proteins. Thus, it is crucial to translate coordinates into a form that the deep learning model can understand. There are two types of successful attempts, one of which is combining coevolutionary information with coordinates [19], and the other is calculating “distance maps” representing the distance between each pair of residues [16]. One strategy to combine structural information with sequence conservation information is spatial filtering [19] (similar to spatial convolution on all the Cα atoms of a protein). Though HHBlits profiles gradually become a dominant feature of sequence conservation [20], its status as the coevolutionary information in spatial filtering has not been evaluated.

In this study, we curated a protein-protein interaction site dataset containing 1742 human proteins reported in DELPHI [9] and generated the 3D structure (PDB files) of these proteins by AlphaFold2. Then, we built a deep learning framework and evaluated the spatial filtering strategy based on the HHblits profiles with various filtering scales, defining the new conservation value of each residue as the mean original conservation value of all residues in its neighborhood. Besides, we implemented the singular value decomposition (SVD) of the distance map and calculated the second structure and relative accessible surface area (rASA) directly from the PDB files generated by AlphFold2 and evaluated these features. Furthermore, we evaluated the effectiveness of these derived features on a membrane protein-metal binding site dataset reported in MPLs-Pred [21] using the precision metric. Computational experimental results showed that all these features were helpful in protein binding site prediction when using deep learning. Finally, we randomly selected the protein P17081 and mutated the 97th residue of its protein sequence from Q to R as an example of a case study. We found that AlphaFold2 might predict shifted 3D structures of some mutated proteins from the corresponding wild-type proteins, and gave a visualized example that the spatial filtering strategy might introduce these structural changes to deep learning models when mutations happen. The datasets and support materials are both available at https://github.com/Liuzhe30/space-hhblits (created on 26 February 2022).

## 2. Materials and Methods

### 2.1. Datasets

We selected 1742 human proteins from the dataset reported in DELPHI [9] to evaluate different coding strategies of structural information on single-sequence PPI site prediction. The sequences were clustered by PSI-CD-HIT [22] with a threshold value of 25%. Next, we extracted the PDB files of these chains from the AlphaFold Protein Structure Database (a set of AlphaFold2 protein structure predictions from DeepMind and EMBL-EBI, version: 2022-02) [23]. These PDB files contained the coordinates of all the atoms of each protein. In this work, the coordinates of alpha carbon atoms (Cα) were taken as the spatial position of residues. Details of the dataset division and binding residues are given in Table 1.

### 2.2. Spatial Filtering with HHBlits

Since the evolutionary information of adjacent residues may improve the prediction accuracy, we defined the neighborhood of each residue as the entirety of all residues with a certain distance centered on the Cα atom itself (Figure 1C). The original evolutionary information was obtained from HHblits, with each protein a 30-dimension matrix [20]. We defined the distance between residues as the Euclidean distance between the coordinates of Cα atoms predicted by AlphaFold2. Then, the conservation value of each residue is replaced by the mean original conservation value of all residues in its neighborhood as a comprehensive spatial regional feature.

### 2.3. Feature Engineering

In this work, we prepared five encoding features to represent the protein sequences: OneHot [24], HHblits profiles [20], SVD [25] of the distance map, secondary structure (SS) [26], and rASA [27]. A sliding window of 31 residues is used for a given protein to predict whether the central residue is a protein binding site.

#### 2.3.1. OneHot and HHblits Profiles

The OneHot and HHblits profiles, which represent the sequential information and sequence conservatism, respectively, are two classic features in sequence-based protein-related prediction tasks [28,29,30]. The HHblits profiles were generated by HHblits [20], and the obtained matrix consisted of 30 dimensions. Note that our main purpose is to evaluate encoding strategies of 3D structure information of proteins. There is no need to test the effectiveness of OneHot and HHblits profiles. Here, we took OneHot and HHblits profiles as the basic feature to ensure the fitting direction of the model.

#### 2.3.2. SVD of the Distance Map

As for encoding protein 3D structures, distance maps of proteins representing the pairwise distance between residues have proved useful in previous research. However, the truncation of over-long proteins is required to adapt the input size of the neural network. It would result in poor scalability of deep learning methods. Singular value decomposition (SVD) could reduce dimension while retaining important components of the original feature as much as possible [25]. Here, we performed SVD on the distance map of the proteins according to the formula:(1)$$A=U\Sigma V^{T}\;$$
where *A* is the matrix to be decomposed, *U* and *V* are the left and right singular matrices, respectively, and *Σ* is the singular value matrix. The distance map was represented by the first *k* vectors of the $V^{T}\;$ matrix,
(2)$$A\prime_{m\times k}=A_{m\times n}V_{n\times k}$$
where *m* is the dimension of the protein sequence, and *n* is equal to m since the distance map is symmetric.

#### 2.3.3. The Secondary Structure

Site-specific mutation studies are designed using the secondary structure (SS) of proteins, which also aids in identifying functional domains [26]. In this study, we assigned each residue’s secondary structure into three categories: helix (H), strand (E), and coil (C). Through PDB files generated by AlphaFold2, we were able to extract the secondary structure labels by the DSSP [31] (merge ‘H’, ‘G’, ‘I’ as ‘H’; merge ‘B’, ‘E’ as ‘E’; merge ‘T’ and others as ‘C’).

#### 2.3.4. The Relative Accessible Surface Area

The relative accessible surface area (rASA) measures the solvent exposure of protein residues [27]. rASA is also reported as a powerful property relevant to the characterization of PPI-binding sites [11]. It can be calculated by the formula:(3)$$rASA=\frac{ASA}{maxASA}\;$$
where *ASA* is the surface area that is accessible to solvents, and *MaxASA* is the residue’s largest conceivable solvent-accessible surface area. Here, the protein sequences were used to predict PDB files using AlphaFold2, and DSSP [31] was used to determine each residue’s ASA value. The rASA feature was then estimated using the MaxASA values that had been reported (Appendix A) [32].

### 2.4. Deep Learning Network and Training Strategy

#### 2.4.1. The Architecture of the Deep Artificial Neural Network

To assess the contributions of each feature, we designed an MLP model with many inputs. The input was fed parallelly into two hidden layers, a dropout layer, a batch normalization layer, and a dense layer. The model outputs the classification results from the final dense layer with a softmax activation after concatenating all branches of various features (Figure 2). It should be noted that only the most basic MLP model was created and utilized for comparison because our goal was to evaluate the validity of features rather than to provide a high-performance prediction tool.

The model was implemented using Keras [33] and TensorFlow [34], and we trained the network on Nvidia GeForce RTX 3090 GPUs. Here, 1024 was chosen as the batch size. Weighted binary cross-entropy was used as the loss function. We set the class weights of binding sites and non-binding sites to 10 and 1, respectively.

#### 2.4.2. Performance Evaluation

Precision was chosen as the assessment parameter in this downstream task because predicting binding sites and non-binding sites are regarded as a classification challenge, and we were more interested in the model’s capacity to identify actual protein binding sites. The formula for calculation of *Precision* is as follows:(4)$$Precision=\frac{TP}{TP+FP}\;$$
where true positive (*TP*) stands for the number of binding sites (residues) that were predicted properly, and false positive (*FP*) stands for those that were predicted wrongly. It is important to note that, in order to make comparisons easier, we employed just one assessment metric. This was done since we only wanted to compare the advantages of various features within the same model, not the predictor’s overall performance.

## 3. Results

### 3.1. Comparison between Various Spatial Filter Radiuses with HHblits

Previous work has proposed spatial neighbor–based PSSM to predict RNA-binding sites using structure-known proteins and proved this new encoding strategy outperformed the standard and smoothed PSSM [19]. However, serving as a conservative profile like PSSM, the HHblits profile has not been evaluated. To explore whether the spatial filter strategy is effective in information fusion when AlphaFold2 serves as the 3D structure predictor, we evaluated the effect of its combination with HHBlits under different filter radiuses (see Table 2). In this experiment, the baseline of the control was defined as the model’s prediction performance trained on the simple splicing of OneHot and the standard HHblits profiles (“0 Å”).

It can be observed that by taking the average performance of the five replicates as the criterion, with the increase in filtering radius, the predicted performance increased first and then decreased compared with the baseline (average precision = 0.276 ± 0.035), reaching a peak at about 7 Å (average precision = 0.630 ± 0.075), probably due to the large size of filters causing information confusion and introducing noise into the model. The experimental results show that the spatial filtering strategy based on HHblits is an effective and acceptable way to encode structural information for deep learning models.

### 3.2. Ablation Study of AlphaFold2-Derived Features on Single-Sequence PPI Site Prediction

We also evaluated the contribution of other features based on the protein structure predicted by AlphaFold2 besides HHblits. This work evaluated the singular value decomposition (SVD) of the distance map, the secondary structure, and the relative accessible surface area (rASA) of each residue. Previous work usually fetched the secondary structure and rASA using sequenced-based prediction tools when the real structures of proteins were unknown [35,36]. Now with AlphaFold2, we have the opportunity to directly calculate these features from the predicted accurate 3D structures as new predicted features different from those predicted by traditional sequence-based tools. For the convenience of calculation and scalability, we adopted eight dimensions and 16 dimensions of features extracted from singular value decomposition of the distance matrix (SVD8 and SVD16).

As shown in Table 3, compared with the baseline (average precision = 0.276 ± 0.035), the introduction of SVD8, SVD16, SS, and rASA all positively affect the prediction of binding sites and non-binding sites. It was observed that the performance with the introduction of rASA is marginal with the precision of 0.594 ± 0.078, which means that the rASA based on the predicted protein structure strongly improves the performance of prediction of PPI sites. This result is consistent with the experimental report that solvent accessibility plays a critical role in PPI site prediction [37]. We noticed that the performance of SVD16 was slightly decreased compared with SVD8, which may be due to the redundant information introduced by the high-dimensional matrix. Since the rotation invariance of deep learning networks hurts their performance when handling numerous uninformative features, the redundant information of the higher-dimensional SVD matrix might cause a decrease in the prediction performance [38].

### 3.3. Evaluation of AlphaFold2-Derived Features on Membrane Protein-Metal Binding Site Prediction

Membrane proteins (MPs) are an important type of protein involved in various crucial biological functions [39]. Locating ligand binding sites and finding the functionally important residues from protein sequences is helpful in understanding their function [40]. Here, we collected 1584 membrane protein chains reported in MPLs-Pred [21] to evaluate the AlphaFold2-derived features on protein-metal binding sites. The sequences were clustered using CD-HIT with a threshold value of 30% [22]. The predicted 3D structure of these proteins was obtained from the AlphaFold Protein Structure Database [23]. In this section, five features were generated for evaluation, namely Space-HHblits (with a radius of 7 Å), SVD8, SVD16, SS, and rASA. The ratio of class weights was set to 3:1 for positive and negative samples, respectively.

As shown in Table 4, compared with the baseline (average precision = 0.470 ± 0.039), the introduction of Space-HHblits (with the radius of 7 Å), SVD8, SVD16, SS, and rASA all improved the prediction of binding sites and non-binding sites. In this task, SVD8 performed best with the precision of 0.627 ± 0.019, while rASA brought a relatively small boost (average precision = 0.591 ± 0.020). Different from the prediction of the PPI binding site, membrane proteins have a special external environment, and the binding site of metal ions may not completely depend on the exposure of residues. As in PPI binding site prediction, the performance of SVD16 was slightly decreased compared with SVD8, indicating that it is the efficiency of features, not dimensionality, that is more useful for prediction.

### 3.4. An Example of a Protein Carrying Variant

Variations in proteins may result in disrupting existing protein interactions or forming new interactions [41]. Prediction tasks, such as estimating the mutations’ effects on PPI, would encounter the challenge of how to teach the model to realize the changes in proteins were caused by mutations [42]. The common solution is to compute the protein sequence conservation before and after the mutation happens. AlphaFold2 used a multiple sequence alignment (MSA) during training [43], which could identify the difference between sequences with changes [44], such as mutational changes; thus, we assume that AlphaFold2 might produce a different protein structure for input sequence with changes based on alignment results from MSA (Appendix A).

In this section, we used a protein chain (UniProt ID: P17081) as an example case to assess whether spatial filtering could efficiently encode the structural effects of mutations. We artificially mutated the 97th residue from Q to R (P17081_Q97R), as shown in Figure 3.

We then produced standard HHblits profiles for P17081 and P17081_Q97R and performed spatial filtering based on their corresponding structures. In spatial filtering, the average value is calculated each time when a neighbor is added to amplify the benefit of smoothing brought by filtering. Next, for standard HHblits and spatially filtered HHblits profiles, we calculated the change rates, *R_HHblits__−__classical_* and *R_HHblits__−__spatial_* using the following formula:(5)$$R_{HHblits}=\frac{\Delta HHblits}{Original\;HHblits}=\frac{Mutated\;HHblits-Original\;HHblits}{Original\;HHblits}\;.$$

As shown in Figure 4, before spatial filtering, the pattern of the change rate of standard HHblits profile was seemingly random, while the feature graph became smooth after filtering. Remarkably, in Figure 4B, the change rate of the 29th residue (Y), instead of the central residue, was shown as a highlighted line. Next, we calculated the Euclidean distance of each residue to the mutated site before and after the mutation happened. We found that the value of the change rate of distance at the 29th residue (Y) was also at its maximum (Figure 4C). This indicated that by taking the mutation position as the reference point, the 29th residue (Y) was the one with the largest spatial position change upon the mutational impact.

The computational experimental results show that AlphaFold2 might have the ability to predict a shifted 3D structure of the mutant protein from the corresponding wild-type protein, and the spatial filtering strategy might have the potential to embed the structural variation information into deep learning models when a mutation happens. This approach provides a new feature of the mutant protein carrying information different from the original protein, which might be helpful in some sequence-based deep learning tasks, such as predicting the effect of mutations on PPIs. Although AlphaFold2 currently is unable to predict all the structural effects of missense mutations correctly, it is conceivable that the incorporation of structure-disrupting mutations experimental data will enable this feature in future versions of protein structure prediction programs [45].

## 4. Discussion

The appearance of AlphaFold2 sheds new light on some tasks that are limited by the need for known protein structures. In this study, we evaluated the effects of spatial filtering, SVD of the distance map, calculating the secondary structure of proteins, and the rASA feature with the help of AlphaFold2 on single-sequence protein binding site prediction. The computational experiment results showed that all these strategies were effective and suitable to encode structural information for deep learning models. These coding strategies with AF2-predicted structures give new predicted features that might be useful for some deep learning-based prediction tasks.

Some limitations can still be found in our work. Firstly, in the computational experimental design, some variables, such as sliding window size, batch size, and model architecture, were not tested. Since we only want to measure the benefit of different features calculated from the PDB files predicted by AlphaFold2, rather than evaluate the overall performance of different deep learning networks, only a simple MLP model was implemented, and the parameters of the model were fixed after simple adjustments based on experience. Moreover, limited by computing cost, we did not evaluate the distance map but reduced its dimension with the help of SVD. Using a sliding window will undoubtedly further reduce the information carried by the distance map, which may lead to the deviation of the original effect of this feature.

It is believed that the emergence of AlphaFold2 brings both opportunities and challenges. In the future, we will continue to explore different coding approaches to AlphaFold2′s outputs and validate them on more downstream tasks (including mutation-containing tasks) with more complicated deep learning networks. Since there are many well-established models for protein and protein complexes structure prediction other than AlphaFold, we will also evaluate various coding strategies of the 3D structure predicted from different predictors in our next version.

## 5. Conclusions

In this study, we evaluated the effects of five processing strategies of coordinates generated by AlphaFold2, namely spatial filtering, the singular value decomposition (SVD) of distance map (8D and 16D), and calculating the secondary structure feature and relative accessible surface area (rASA) feature of proteins. The computational experiment results showed that all these strategies were effective and suitable to encode structural information for deep learning models. These strategies generate new predicted features that might be useful in some deep learning-based tasks, such as predicting the binding site of proteins from primary sequences.

## Figures and Tables

**Figure 1 biology-11-01454-f001:**
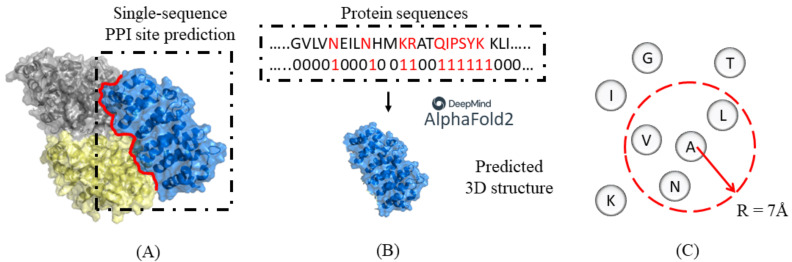
(**A**) The prediction task of single-sequence PPI site prediction. (**B**) The figure showed the protein 3D structure predicted from primary protein sequences using AlphaFold2. (**C**) In the spatial filtering strategy, “neighborhood” is defined as the entirety of all residues in a space with a certain distance centered on the Cα atom of the current residue.

**Figure 2 biology-11-01454-f002:**
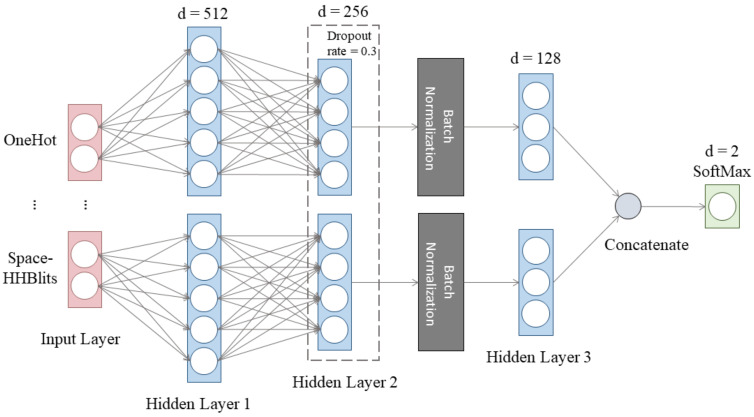
The structure of the MLP model.

**Figure 3 biology-11-01454-f003:**
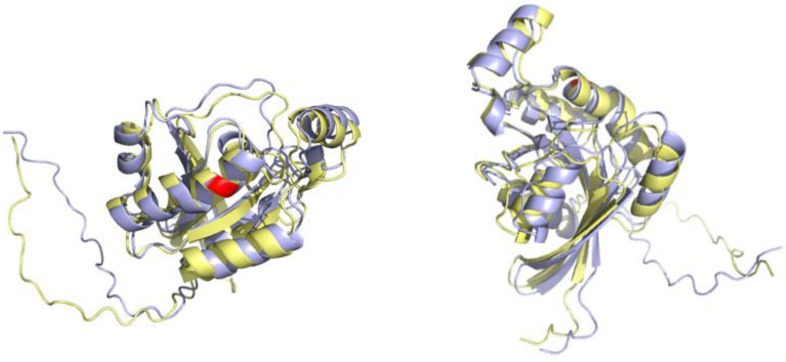
The 3D visualization of UniProt ID: P17081 (colored in yellow) and P17081_Q97R (colored in blue) using PYMOL. The mutated residue is colored red. The folded structure of mutant protein predicted by AlphaFold2 is roughly identical to the pre-mutant protein, except for some minor shifts. The RMSD between the structure of P17081 and P17081_Q97R is 0.046 Å calculated by PYMOL.

**Figure 4 biology-11-01454-f004:**
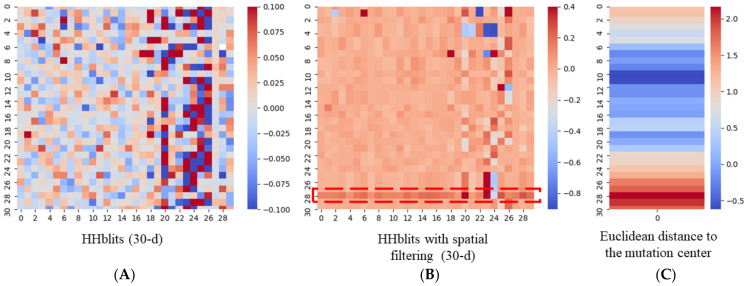
The visualization of the change rate of HHblits and Euclidean distance to the mutated residue before and after mutation. Taking the mutant residue as the center, we simulated the operation of the sliding window and clipped the sequence into a fragment with a length of 31 residues. The 16th position represents the mutant residue. (**A**) Heatmap of the change rate of classical HHblits profiles. (**B**) Heatmap of the change rate of HHblits with spatial filtering (radius = 7 Å). The highlighted area represents the 29th residue (Y) of the current window. (**C**) Heatmap of the change rate of the Euclidean distance to the mutation residue.

**Table 1 biology-11-01454-t001:** The datasets for training, validation, and testing are shown in the table.

Dataset	Proteins	Residues			% Binding out of Total
		Total	Binding	Non-Binding	
Training	1682	970,997	66,143	904,854	6.8119
Validation	30	16,893	1607	15,286	9.5128
Test	30	20,051	1460	18,591	7.2814
Total	1741	1,007,941	69,210	938,731	6.8665

**Table 2 biology-11-01454-t002:** Comparison of different spatial filter radiuses. Five independent replicates were performed for each radius to eliminate random errors.

	E1	E2	E3	E4	E5	Average	STD
0 Å (no filtering)	0.344	0.264	0.250	0.246	0.276	0.276	0.035
3 Å	0.244	0.245	0.447	0.314	0.234	0.297	0.080
5 Å	0.282	0.275	0.231	0.196	0.427	0.282	0.079
6 Å	0.299	0.503	0.399	0.515	0.406	0.424	0.088
7 Å	**0.623**	**0.743**	0.508	**0.656**	**0.618**	**0.630**	0.075
8 Å	0.545	0.472	**0.646**	0.603	0.544	0.562	0.066
9 Å	0.575	0.482	0.369	0.450	0.378	0.451	0.075
11 Å	0.368	0.445	0.421	0.616	0.451	0.460	0.083

Bold fonts are the best computational experimental results. E1–E5 represents the precision metric of five independent replicates on the test set.

**Table 3 biology-11-01454-t003:** Ablation study of AlphaFold2-derived features on single-sequence PPI site prediction.

	E1	E2	E3	E4	E5	Average	STD
Onehot + HHblits (Baseline)	0.344	0.264	0.250	0.246	0.276	0.276	**0.035**
Onehot + HHblits + SVD8	**0.540**	0.359	0.453	0.587	0.551	0.498	0.092
Onehot + HHblits + SVD16	0.459	0.505	0.460	0.371	0.396	0.438	0.054
Onehot + HHblits + SS	0.510	0.348	0.471	**0.594**	0.585	0.502	0.100
Onehot + HHblits + rASA	0.479	**0.603**	**0.673**	0.536	**0.681**	**0.594**	0.078

Bold fonts are the best computational experimental results. E1–E5 represents the precision metric of five independent replicates on the test set.

**Table 4 biology-11-01454-t004:** Evaluation of features on membrane protein-metal binding site prediction.

	E1	E2	E3	E4	E5	Average	STD
Onehot + HHblits (Baseline)	0.528	0.428	0.505	0.442	0.446	0.470	0.039
Onehot + Space-HHblits (7 Å)	**0.645**	0.601	0.602	0.620	0.588	0.611	0.020
Onehot + HHblits + SVD8	0.601	**0.630**	**0.640**	**0.653**	**0.610**	**0.627**	0.019
Onehot + HHblits + SVD16	0.600	0.629	0.639	0.606	0.574	0.610	0.023
Onehot + HHblits + SS	0.643	0.620	0.612	0.597	0.635	0.621	**0.016**
Onehot + HHblits + rASA	0.556	0.602	0.615	0.598	0.586	0.591	0.020

Bold fonts represent the best computational experimental results. E1–E5 represents the precision metric of five independent replicates on the test set.

## Data Availability

All data analyzed in this study are curated from the public domain.

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
