# Peer review of "Evaluation of the Effectiveness of Derived Features of AlphaFold2 on Single-Sequence Protein Binding Site Prediction"

_biology, 2022, doi:10.3390/biology11101454_

Round 1

Reviewer 1 Report

Dear Authors,

Thank you for the chance to review the manuscript entitled “Evaluation of the Effectiveness of Derived Features of AlphaFold2 on Single-sequence Protein Binding Site Prediction”. This article presents helpful insights into the implementation of new/optimized features (e.g. residue surface area) to improve protein structure prediction models, such as AlphaFold2. In summary, the article reads very well and the introduction, methods, and results are all clearly described.

Minor points for suggestion below:

1) Line 172 … seems like a word is missing between “However,” and “serves”

2) I am not sure how section “3.4 An Example of a Protein Carrying Variant” is related to protein binding site prediction as mentioned in the title. Please elaborate or include information about predicting changes in protein-protein interactions.

3) How do these findings compare to similar ideas of improving prediction models with new features? Would it be possible to reference other such improvements in the Discussion section? Also, please discuss the findings in the Discussion section. The limitations are very nicely described, but the results are missing in this section from what I can see.

Congratulations on the great work!

Author Response

Response to reviewers

Reviewer #1:

Thank you for the chance to review the manuscript entitled “Evaluation of the Effectiveness of Derived Features of AlphaFold2 on Single-sequence Protein Binding Site Prediction”. This article presents helpful insights into the implementation of new/optimized features (e.g. residue surface area) to improve protein structure prediction models, such as AlphaFold2. In summary, the article reads very well and the introduction, methods, and results are all clearly described.

Response: We thank the reviewer for the positive comments about our manuscript and the work presented therein. We have carefully considered all comments in light of improving the quality of the manuscript. Following the reviewer’s comments, we have made corresponding modifications to our revised manuscript.

1) Line 172 … seems like a word is missing between “However,” and “serves”

Response: We apologize for this mistake, and we have rephased the sentence as follows:

In ‘3.1. Comparison between various spatial filter radiuses with HHblits’ section: “However, serving as a conservative profile like PSSM, the HHblits profile has not been evaluated.”

2) I am not sure how section “3.4 An Example of a Protein Carrying Variant” is related to protein binding site prediction as mentioned in the title. Please elaborate or include information about predicting changes in protein-protein interactions.

Response: We thank the reviewer for this comment and suggestion. And we recognized this as one of the limitations of our current manuscript. We have not evaluated the coding strategies in the prediction tasks, such as predicting changes in protein-protein interactions, since the performance of AlphaFold2 on mutant datasets has not been extensively validated. We used this case study to illustrate that these coding strategies might have more applications. We are considering evaluating the coding strategies mentioned in the current manuscript on mutation-related downstream tasks in the next version. Thus, we included the following sentence about this potential application in the discussion in the revised manuscript:

In the ‘Discussion’ section: “In the future, we will continue to explore different coding approaches to AlphaFold2’s outputs and validate them on more downstream tasks (including mutation-containing tasks) with more complicated deep learning networks.”

3) How do these findings compare to similar ideas of improving prediction models with new features? Would it be possible to reference other such improvements in the Discussion section? Also, please discuss the findings in the Discussion section. The limitations are very nicely described, but the results are missing in this section from what I can see.

Response: We thank the reviewer for this suggestion. The findings of our paper can be presented as the similar ideas of improving prediction models with new features carrying structural information of proteins. Precisely, these coding strategies will not only introduce the predicted structural information into the deep learning network but also can be taken as new features that might help improve deep learning models. We have added this improvement into the ‘Discussion’ section and discussed the findings of this paper:

In the ‘Discussion’ section: “The computational experiment results showed that all these strategies were effective and suitable encode structural information for deep learning models. These coding strategies with AF2-predicted structures give new predicted features that might be useful for some deep learning-based prediction tasks.”

Reviewer 2 Report

The authors presented five strategies for deriving structural information from protein conformations generated with novel AlphaFold2 software. The extracted features are suggested as useful in training the machine-learned model for protein binding site prediction.

The presented work includes a simple, not-optimized artificial intelligence model built with TensorFlow and Keras. Using the model and 'precision parameter' as the only metric of prediction performance, the authors assessed that all strategies proposed were effective and suitable to encode structural information for deep learning models.

However, the idea behind the computational experiment seems interesting, but the implementation and robustness of testing require improvement. The source code of the model is not provided as well as a detailed description of the protocol applied. Therefore, the practical applicability, usefulness, and educational value for the users are minor.

Also, the authors presented an example of capturing the structural changes resulting from the point mutation. Based on the single case study they also provided a strong statement that "AlphaFold2 allows us to directly capture alterations caused by mutations at the level of 3D structure". An extensive study on a large dataset would be needed to confirm the veracity of this statement, and that may go beyond the scope of this work. However, for the hypothesis itself, it would be necessary to provide some (3-5) examples as a brief summary in the manuscript and detailed results in supplementary materials.

Lastly, why were the authors limited to studying the conformations generated with AlphaFold2 and how are they sure that these structures are correct? The results obtained should be compared at least for several cases (when available) with reference results for known experimental protein structures. Also, there is much well-established software for protein and protein complexes structure prediction other than AlphaFold, so do the authors think their approach will be suitable to derive structural information from atomic coordinates obtained with those tools? Why limit the applicability of the protocol to results from one tool? Comparative work would be more objective and consolidate knowledge about the general correctness and usefulness of predicted protein structures. 

Below, please find some detailed comments:

- lines 43-44: soften or precise the statement of "lack of accurately measured protein crystal structures"

- line 47: the CASP results showed that the accuracy is not high enough for a third of AlphaFold2 predictions, so I may suggest altering the sentence to "...predict the 3D structure of some proteins with remarkable accuracy... "

- line 54: please briefly explain in the manuscript why directly feeding coordinates into the model is not an ideal choice 

- line 59: please briefly explain what spatial filtering is and what techniques you used

- lines 59-60: please rewrite the sentence, HHBlits is a software

- line 68: please briefly explain how did you evaluate the features

- lines 72-76: please provide the conclusion from at least several case studies

- line 87: did you run the AF2 predictions or took the structures from the AlphaFold database or other resources of pre-calculated conformations

- line 99: missing close sentence punctuation

- lines 98-99: please briefly explain why did you use the mean conservation value of all residues in its neighborhood

- line 102: please provide references for tools and approaches

- lines 114-115: please use precise terminology; contact maps are distance maps filtered by the distance cutoff assumed for contacts; distance maps provide a pairwise distance for each-of-each  

- lines 129-130: please use precise terminology: secondary structure can not be divided, you can assign secondary structure in 3-letter notation or provide the probability of each type of secondary structure for a given residue

- line 132: since the DSSP algorithm provides secondary structure assignment in 8-letter notation, please specify how you simplified it into the 3-letter notation

- line 139:  please use precise terminology: protein sequences can not be converted into the PDB

- lines 141-142: the reference you provided compared several scales of maxASA, including their novel measure; please be precise for the readers which scale you use and from which table in the reference

- Tables 2, 3, and 4: what are the values in the table? is that model's prediction performance or precision parameter? what are the units or allowed range of values? how was it calculated and on how many proteins (is it average)? please provide more specific description in the table title

- line 187, 282, etc.: please use rather a computational experiment term instead of "experimental results" or "experimental design" since they suggest that there was some wet-experimental work done in this study

- lines 196-197: if there is no known structure in the Protein Data Bank database as a reference, how do you know the predicted 3D structure with AF2 is correct?

- lines 220-221: again, did you use AF2 to predict structures or downloaded the pre-computed structures from the repository

- lines 243-244: please soften the statement because based on the so few examples shown it is hard to assess that for sure

- Figure 3: could you color one pair of structures (e.g., right panel ) using AlphaFold confidence error? for example, you can use two different color scales for a wild type (red-yellow) and mutant (blue-green); that should help to assess if the structural changes result from the amino acid substitution or maybe the reason is low confidence of the prediction

- paragraph starting from line 260: please explain in the text of the manuscript what is residue 29th; for clarity, it should be in the text not only in the caption of Figure 4

- discussion section: the statement is too general, please narrow the scope of applicability to provide a realistic and reliable overview; also, consider discussion of potential usage combined with protein structures obtained from different tools

- line 292: the authors mention five features, but list only four of them; please add in the brackets the techniques used for spatial filtering

- conclusion section: more detailed and comparative analysis of precision provided in the tables may bring additional value to the manuscript; please list the potential practical applications of using the proposed approach

Author Response

Please see "Response to the Review 2" in the attachment.

Reviewer 3 Report

Manuscript titled “Evaluation of the Effectiveness of Derived Features of AlphaFold2 on Single-sequence Protein Binding Site Prediction” by Zhe Liu et. al., describes how to process and encode the AlphaFold2 predicted results effectively to improve the performance of dowstream tasks. They tested the effects of five processing strategies of coordinates on two single-sequence protein binding site prediction tasks. They claim that all strategies they employed were suitable and effective methods to encode structural information for deep learning models. In summary, the authors claim that their study provides new insight into the downstream tasks of protein-molecule interaction prediction. Overall, this is a nice study (with detailed methodological explanation) to encode the structural insights for deep learing models. It will be great if authors can provide couple of more examples to test their strategies (at least to test the mutation study) as AlphaFold is still an emerging field to study protein-molecule interaction prediction. I have following comments for the authors:

1. In Table 2 (Comparison of different spatial filter radiuses), Five independent replicates were performed for each radius. Authors describe that with the increase of filtering radius, the predicted performance increased first and then decreased compared with the baseline. But this observation is not consistent with all the replicates (for example, E3, E4). Can authors explain this?

2.     The performance of SVD16 is seems slightly decreased compared with SVD8, and authors claim that it could be due to the redundant information introduced by the high-dimensional matrix. Can authors elaborate this?

3.     What is the RMSD of the mutant protein structure (Fig. 3) predicted by AlphaFold2 compared to the pre-mutant protein?

4.     One of the major parts missing in this study is lack of evaluation of distance map. By any chance, can authors include this in the current version?

5.     General comment: Can authors speculate (using any of their strategies) how AlphaFold2 predicts the disorder/flexible regions present in protein structure? 

Author Response

Please see "Response to the Review 3" in the attachment.

Round 2

Reviewer 2 Report

Most of my key comments have been clarified and completed. The manuscript now presents a much higher quality of content and should be more useful to the reader.

Appreciation to the authors for the work they put into improving their scientific report. 

Author Response

We are grateful for the reviewer's guidance and assistance in revising our manuscript and thank the reviewer for the positive comments about our manuscript!